# Fast and accurate joint inference of coancestry parameters for populations and/ or individuals

**Tristan Mary-Huard** [1,2]*, **David Balding**[3]

**1** MIA-Paris, INRAE, AgroParisTech, Université Paris-Saclay, Palaiseau, France, **2** Université Paris-Saclay, INRAE, CNRS, AgroParisTech, Génétique Quantitative et Evolution–Le Moulon, Gif-sur-Yvette, France, **3** Melbourne Integrative Genomics, School of BioSciences and School of Mathematics & Statistics, University of Melbourne, Parkville, Victoria, Australia

* tristan.mary-huard@agroparistech.fr

**Data Availability Statement:** All data are fully available without restriction here: https://ftp. 1000genomes.ebi.ac.uk/vol1/ftp/release/ 20130502. Functions to compute Fst estimates and perform tree inference can be found in the R

## Abstract

We introduce a fast, new algorithm for inferring from allele count data the $F_{ST}$ parameters describing genetic distances among a set of populations and/or unrelated diploid individuals, and a tree with branch lengths corresponding to $F_{ST}$ values. The tree can reflect historical processes of splitting and divergence, but seeks to represent the actual genetic variance as accurately as possible with a tree structure. We generalise two major approaches to defining $F_{ST}$, via correlations and mismatch probabilities of sampled allele pairs, which measure shared and non-shared components of genetic variance. A diploid individual can be treated as a population of two gametes, which allows inference of coancestry coefficients for individuals as well as for populations, or a combination of the two. A simulation study illustrates that our fast method-of-moments estimation of $F_{ST}$ values, simultaneously for multiple populations/individuals, gains statistical efficiency over pairwise approaches when the population structure is close to tree-like. We apply our approach to genome-wide genotypes from the 26 worldwide human populations of the 1000 Genomes Project. We first analyse at the population level, then a subset of individuals and in a final analysis we pool individuals from the more homogeneous populations. This flexible analysis approach gives advantages over traditional approaches to population structure/coancestry, including visual and quantitative assessments of long-standing questions about the relative magnitudes of within- and between-population genetic differences.

## Author summary

We propose new ways to measure, and visualise in a tree, the genetic distances among a set of populations using allele frequency data. The two genomes within a diploid individual can be treated as a small population, which allows a flexible framework for investigating genetic variation within and between populations. Genetic structure is represented in a best-fitting tree with nodes corresponding either to populations or to individuals. This permits both homogeneous populations and genetically diverse individuals, for example

package HFst, available at https://github.com/tmaryhuard/HFst along with a short tutorial on population analyses using chromosomes 21 and 22.

**Funding:** This research was partially supported by grant DP210102168 from the Australian Research Council to DB, and by the "Investissement d'Avenir" project (Amaizing, ANR-10-BTBR-0001) to TMH. The funders had no role in study design, data collection and analysis, decision to publish, or preparation of the manuscript.

**Competing interests:** The authors declare that no competing interests exist.

due to admixture, to be represented efficiently and informatively. We first generalise the long-established measure of genetic distance, $F_{ST}$, to tree-structured populations and individuals, deriving two measures for each pair of populations, corresponding to their shared and non-shared genetic variation. We show using a simulation study that our tree-based estimators can be more efficient than current pairwise estimators, and we illustrate the potential for novel ways to explore and visualise genetic variation within and between populations using a worldwide human genetic dataset.

## Introduction

$F_{ST}$ is a measure of between-population genetic distance introduced in the seminal work of [1]. Several definitions have been proposed, for example in terms of correlations of alleles sampled from populations, relative to an actual or hypothetical reference population, or in terms of average mismatch probabilities for pairs of alleles from the same population, and from different populations. Different underlying definitions have complicated comparisons of the many $F_{ST}$ estimators that have been proposed. These include sum-of-squares estimators in a components-of-variance framework [2], and maximum likelihood estimation based on the variance parameter of the multinomial-Dirichlet distribution (beta-binomial for diallelic markers) [3]. Many method-of-moments (MoM) approaches have been proposed [4–8], based on statistics that measure matching alleles or, equivalently, mismatches (often referred to as expected heterozygosity). The MoM estimators are generally simple and computationally fast, suitable for the very large numbers of single-nucleotide variants (SNV) that are now available. With such large SNV datasets, estimators can be precise and so differences in definitions can be important. $F_{ST}$ estimates for pairs of worldwide human populations have differed by almost a factor of two, due to sensitivity to the minor allele frequency (MAF) [7].

Currently, researchers with multi-population data typically apply a standard estimator separately for each pair of populations. Recent advances [8, 9], following earlier suggestions [10, 11], have added flexibility through integrating the analyses of individuals and populations. Here we propose fast and statistically-efficient MoM estimation of $F_{ST}$, simultaneously for multiple populations and/or individuals, by inferring a tree of ancestral populations with branch lengths that can be used to compute shared and unshared components of allele-frequency variance.

Unrelated diploid individuals can be treated as populations of two gametes. While accurate tree inference with all tip nodes corresponding to individuals is infeasible for large sample sizes, a hybrid approach can be used with some tips corresponding to homogeneous populations and others to individuals with greater genetic diversity, perhaps due to admixture. A flexible modelling framework using a sequence of analyses can be employed to converge on a best-fitting representation of population structure. Here, we do not model the effects of linkage, and so we only consider unrelated individuals (no very recent shared ancestors).

Inbreeding is one of the evolutionary forces that contributes to genetic distance. To introduce our new approach simply, we do not distinguish the effects of inbreeding in the main text, but in S3 Text we outline a more computationally-demanding extension to jointly estimate inbreeding and coancestry parameters.

While the tree typically reflects evolutionary history, it primarily provides a visual representation of the actual genetic variance inferred from observed allele frequencies. Many authors interpret coancestry parameters in terms of identity-by-descent (IBD) probabilities [9, 12, 13]. The IBD concept is popular and allows an intuitive language, but can be problematic [14]

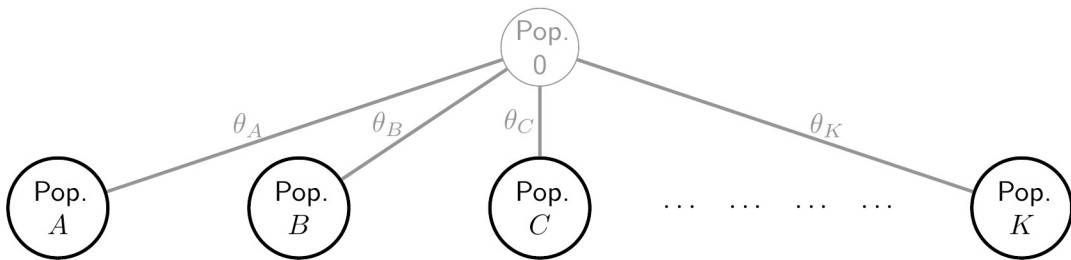

**Fig 1. The independent-descent population model.** The ancestral population (Population 0) is unobserved. Allele count data are available from each of the other populations, which are assumed to have evolved independently from Population 0, with the level of divergence reflected in the $\theta$ values.

because there is a common ancestor at each genome site and different approaches are used to convert the continuous time-since-common-ancestor into a binary IBD state. These include current or ancestral reference populations [8, 9, 13] or mutation events [11, 15]. Our coancestry parameters describe components of allele frequency variance, but our framework is similar to the IBD-centred approach of [9] in that reference allele frequencies are assumed to be those of the most recent population ancestral to the sampled populations. It follows that $F_{ST} \geq 0$ here, as in [9], whereas other approaches allow $F_{ST} < 0$ [7, 8].

We first generalise to multiple populations the correlation and mismatch definitions, denoted $F_{ST}^W$ and $F_{ST}^H$, respectively, the superscripts referring to the seminal authors Weir/Wright and Hudson. For tree-structured populations, $F_{ST}^W$ and $F_{ST}^H$ capture complementary aspects of population structure, corresponding to the lengths of shared and non-shared branches between two populations and the inferred ancestral population. For example, in Fig 1 there are no shared branches and all between-population $F_{ST}^W$ values are zero, but in Fig 2 Populations $C$ and $D$ share Branches 2 and 3, whose lengths contribute to $F_{ST}^W(CD)$, while the lengths of non-shared Branches $C$ and $D$ contribute to the value of $F_{ST}^H(CD)$. Our inferred trees are binary, but since zero branch lengths are allowed, more general tree structures are possible.

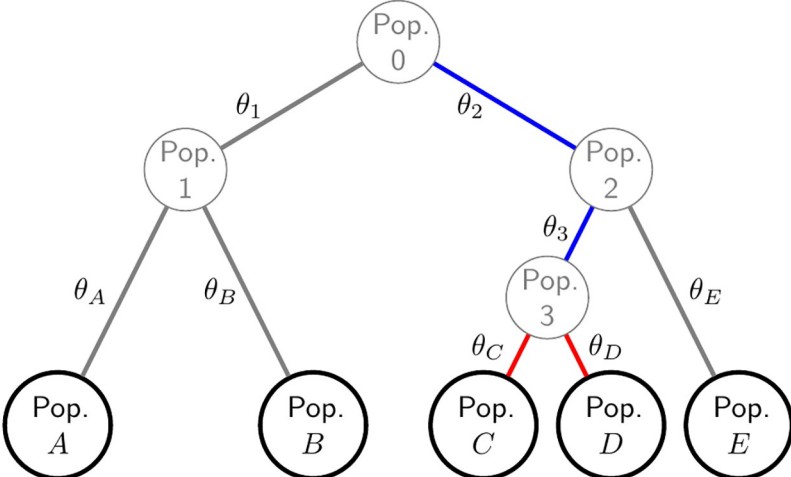

**Fig 2. Tree-structured populations.** Global ancestral Population 0 and intermediate ancestral populations 1 to 3 are unobserved, while allele frequency data is available for populations $A$ to $E$. The genetic differences among the populations are described by the branch lengths, denoted $\theta$. The branches whose lengths contribute to the computation of $F_{ST}^W(CD)$ and $F_{ST}^H(CD)$ are highlighted in blue and red, respectively.

In a simulation study based on a tree-like population structure, we find a small performance advantage of our novel tree-based estimator of $F_{ST}^H$ over the pairwise estimator [7] in most comparisons. Indeed, similar to [6, 8] our estimator draws information from all sampled populations, not only the two being compared. We also simulate a non-tree-like population structure that includes admixing of two highly-diverged populations, finding that the performance advantage is lost for large admixture proportions, but we retain very high correlation of estimates with true $F_{ST}^H$ values. We also analyse the 1000 Genomes Project data and obtain a tree-based representation of the genetic variation among the 26 populations that reveals important insights, even though highly-admixed populations are included. We further investigate 6 of these populations using individual coancestry coefficients, contrasting visually and quantitatively the within- and between-population genetic differences and showing how admixed individuals can be identified and analysed appropriately.

## Materials and methods

### Statistical model and definitions of $F_{ST}$ in the classical setting

Assuming the independent-descent population model of [2] (see Fig 1), write $p$ for the (unknown) reference allele frequency in Population 0 at a given locus, while $p_k$ denotes its value in Population $k$. We assume:

$$\begin{aligned} \mathbb{E}[p_k|p] &= p \\ \mathbb{V}\mathrm{ar}[p_k|p] &= \theta_k p(1-p) \\ \mathbb{C}\mathrm{ov}[p_k, p_{k'}|p] &= 0 \qquad k \neq k' \end{aligned} \tag{1}$$

where $\theta_k \in [0, 1]$ (we use $[a, b]$ and $(a, b)$ for closed and open intervals, respectively, the former including the boundaries $a$ and $b$). Let $x_k, y_{k'} \in \{0, 1\}$ be indicators of the reference allele for random allele draws at the locus in populations $k$ and $k'$. We assume $\mathbb{E}[x_k|p_k] = p_k$, $\mathbb{E}[y_{k'}|p_{k'}] = p_{k'}$ and $\mathbb{C}\mathrm{ov}[x_k, y_{k'}|p_k, p_{k'}] = 0$. Following [6], and close to [1], we define $F_{ST}^W(kk') = \mathbb{C}\mathrm{or}[x_k, y_{k'}|p]$ and from (1) we have

$$F_{ST}^W(kk') = \begin{cases} 0 & \text{if } k \neq k' \\ \theta_k & \text{if } k = k' \end{cases}, \tag{2}$$

In the case $k = k'$, we will write $k$ in place of $kk'$. Correlations can in general be negative, and $F_{ST} < 0$ could arise if the reference population were not ancestral to those sampled [8]. We also define the population analogue of the Hudson estimator [5, 7] as

$$F_{ST}^H(kk') = 1 - \frac{\mathbb{P}[x_k \neq y_k] + \mathbb{P}[x_{k'} \neq y_{k'}]}{2\mathbb{P}[x_k \neq y_{k'}]} . \tag{3}$$

for $k \neq k'$, with $F_{ST}^H(k) = 0$. Here $x_k$ and $y_k$ are indicators for distinct alleles drawn from population $k$. Eq (3) can be viewed as a special case of the multi-population estimator of [8].

Whereas $F_{ST}^W$ is a correlation of sampled alleles, $F_{ST}^H$ is based on mismatch probabilities, or expected heterozygosity, within and between two sampled populations. Under the independent-descent model, (1) leads to $\mathbb{P}[x_k \neq y_{k'}] = p(1-p)$ and $\mathbb{P}[x_k \neq y_k] = (1-\theta_k)p(1-p)$, so that

$$2F_{ST}^H(kk') = \theta_k + \theta_{k'} = F_{ST}^W(k) + F_{ST}^W(k') . \tag{4}$$

While $F_{ST}^W(k)$ measures the divergence of Population $k$ from Population 0, $F_{ST}^H(kk')$ is the average divergence of Populations $k$ and $k'$. The value of $\theta_k$ can be affected by drift, mutation, migration, inbreeding and selection between Populations 0 and $k$. We will see below that in more complex multi-population settings $F_{ST}^W(kk')$ and $F_{ST}^H(kk')$ measure, respectively, shared and non-shared genetic variation in populations $k$ and $k'$.

## The pairwise estimator $\hat{F}_{ST}^H$

At a given locus, the pairwise estimator [7] is $\hat{F}_{ST}^H(kk') = N_{kk'}/D_{kk'}$, where

$$N_{kk'} = (\hat{p}_k - \hat{p}_{k'})^2 - \frac{\hat{p}_k(1-\hat{p}_k)}{n_k - 1} - \frac{\hat{p}_{k'}(1-\hat{p}_{k'})}{n_{k'} - 1} \tag{5}$$

$$\begin{aligned} D_{kk'} &= N_{kk'} + \frac{n_k}{n_k - 1}\hat{p}_k(1-\hat{p}_k) + \frac{n_{k'}}{n_{k'} - 1}\hat{p}_{k'}(1-\hat{p}_{k'}) \\ &= \hat{p}_k(1-\hat{p}_{k'}) + \hat{p}_{k'}(1-\hat{p}_k) \end{aligned} \tag{6}$$

and $n_k$ is the number of gametes sampled in population $k$, while $\hat{p}_k$ is the sample allele frequency. By expanding $(\hat{p}_k - \hat{p}_{k'})^2$, both $N_{kk'}$ and $D_{kk'}$ can be expressed as sums of terms of the form $\hat{p}_k(1-\hat{p}_{k'})$ which for $k \neq k'$ is an unbiased estimator of $p_k(1-p_{k'})$. Assuming the conditional moments (1) we obtain

$$\begin{aligned} \mathbb{E}[N_{kk'}|p] &= (\theta_k + \theta_{k'})p(1-p) \\ \mathbb{E}[D_{kk'}|p] &= 2p(1-p) \ . \end{aligned}$$

From (4), we see that the ratio of the above two expectations is $F_{ST}^H(kk')$ and so, provided that $D_{kk'}$ has a low coefficient of variation which is typically the case in practice, $\hat{F}_{ST}^H(kk')$ is approximately unbiased for $F_{ST}^H(kk')$.

$F_{ST}$ can vary over SNVs, due to locus-specific effects of selection or mutation. In humans, there are relatively few strong outlier SNVs, and these can be removed prior to analysis if required, so that locus-specific selection and mutation effects are often ignored for genome-wide inferences of $F_{ST}$. The multi-locus $\hat{F}_{ST}^H$ is defined by summing numerator and denominator over SNVs:

$$\hat{F}_{ST}^H(kk') = \frac{\sum N_{kk'}}{\sum D_{kk'}} \ . \tag{7}$$

An unbiased alternative is to average the ratios over SNVs but, as previous authors have noted [2, 7, 9], the increased precision of the ratio of averages (7) more than offsets the small bias introduced.

## Tree-structured population setting

We generalise the independent-descent population model to tree-structured populations, see Fig 2 for an example. In place of (1) we now assume

$$\begin{aligned} \mathbb{E}[p_k|p_A] &= p_A \\ \mathbb{V}\mathrm{ar}[p_k|p_A] &= \theta_k p_A(1-p_A) \\ \mathbb{C}\mathrm{ov}[p_k, p_{k'}|p_M] &= 0 \qquad k \neq k' \end{aligned} \tag{8}$$

**Table 1. Notations for population trees.** MRCA = most recent common ancestor. The label of a population also refers to the branch above that population. Intuitively, $\mathcal{Q}(kk')$ denotes the set of "shared" branches that are ancestral to both Population $k$ and Population $k'$, and $\mathcal{R}(kk')$ denotes the set of "non-shared" branches that are ancestral to $k$ but not $k'$.

| Notation | Definition | Examples from Fig 2 |
|---|---|---|
| $M(kk')$ | MRCA of $k$ and $k'$ | $M(CD) = 3$ |
| $\mathcal{P}(k)$ | tree path from 0 to $k$ | $\mathcal{P}(A) = \{1, A\}; \mathcal{P}(C) = \{2, 3, C\}; \mathcal{P}(E) = \{2, E\}$ |
| $\mathcal{Q}(kk')$ | $\mathcal{P}(k) \cap \mathcal{P}(k')$ | $\mathcal{Q}(k) = \mathcal{P}(k); \mathcal{Q}(CE) = \mathcal{Q}(EC) = \{2\}$ |
| $\mathcal{R}(kk')$ | $\mathcal{P}(k) \setminus \mathcal{P}(k')$ | $\mathcal{R}(k) = \emptyset; \mathcal{R}(CE) = \{3, C\}; \mathcal{R}(EC) = \{E\}$ |

where $A = A(k)$ denotes the parent population of Population $k$, and $M = M(kk')$ is the population that is the most recent common ancestor of $k$ and $k'$.

In Theorem 1 (see Results), we extend the definitions of $F_{ST}^W$ and $F_{ST}^H$, and express them as functions of the $\theta_k$. Here we consider the problem of inferring these parameters, along with the tree structure. Similar to the pairwise estimator, the inference procedure will be based on terms of the form $\hat{p}_k(1 - \hat{p}_{k'})$. Let

$$S_{kk'} = \frac{2n_k}{n_k - 1} \frac{\sum N_k}{\sum D_{kk'}} \tag{9}$$

where $N_k = \hat{p}_k(1 - \hat{p}_k)$ and $D_{kk'}$ is defined at (6). The two summations in (9) are over the same set of SNVs, but any SNV that is monomorphic in $k$ and $k'$ combined does not contribute to either sum and hence does not affect $S_{kk'}$. While the fraction of monomorphic sites can be informative about $\theta$ values, data quality issues make it difficult to use this information in real datasets and it is ignored by our estimators.

To understand the statistic $S_{kk'}$ intuitively, assume $n_k$ large so that $n_k/(n_k - 1)$ can be neglected. Then $S_{kk'} = 0$ precisely when $\hat{p}_k$ is either 0 or 1, while $S_{kk'} = 1$ if and only if $\hat{p}_k = \hat{p}_{k'}$. $S_{kk'}$ is undefined if both $\hat{p}_k \in \{0, 1\}$ and $\hat{p}_{k'} \in \{0, 1\}$. For other values of $\hat{p}_k$ and $\hat{p}_{k'}$, we have $0 < S_{kk'} < 1$ with $S_{kk'}$ tending to decrease as the difference between $\hat{p}_k$ and $\hat{p}_{k'}$ increases.

**Proposition 1**. *For $k \neq k'$,*

$$\mathbb{E}[S_{kk'}] \approx 1 - \mathbb{C}\text{or}[x_k, y_k | p_M] = \prod_{q \in \mathcal{R}(kk')} (1 - \theta_q) \tag{10}$$

*where $\mathcal{R}$ is defined in Table 1.*

For the proof, see S1 Text. Proposition 1 states that $1 - S_{kk'}$ is an unbiased estimator of the correlation of two alleles drawn from Population $k$ given the allele frequencies in $M(kk')$. It motivates the following logarithmic least-squares estimation procedure for the $\theta_k$.

## Fast inference of the $\theta_k$ and the population tree

For $K$ populations, and noting that $S_{kk'} \neq S_{k'k}$, there are $K(K - 1)$ values of $S_{kk'}$ available to estimate the $2(K - 1)$ values of $\theta$. Write $\beta_q = \log(1 - \theta_q)$. We estimate $\beta$. the vector of all $\beta_q$ coefficients, by solving:

$$\hat{\beta} = \underset{\beta}{\arg\min} \ \xi \qquad \text{where} \qquad \xi = \sum_{k \neq k'} \left( \log(S_{kk'}) - \sum_{q \in \mathcal{R}(kk')} \beta_q \right)^2 \tag{11}$$

subject to $\beta_q \leq 0$ since $\theta_q \geq 0$.

We propose a fast algorithm to jointly infer the tree topology and $\theta$ values. Restricting the search to binary trees ensures that any two trees with $K$ tip nodes have the same number of $\theta$

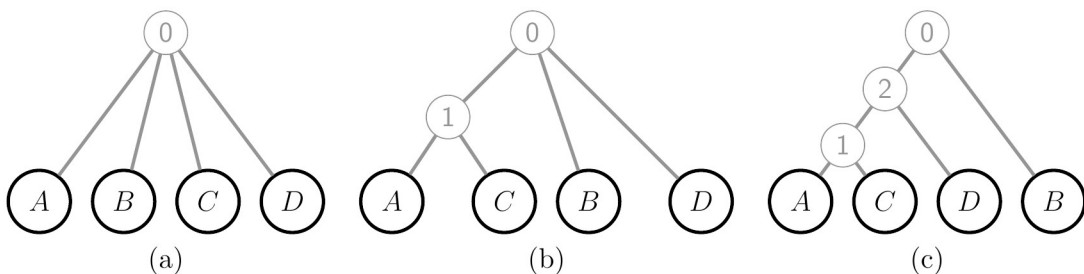

**Fig 3. Hierarchical clustering to infer a binary tree with $K = 4$ sampled populations.** (a) The starting tree corresponds to the independent-descent model, and has all populations directly connected to ancestral Population 0. (b) We identify the pair of populations (here $AC$) such that an intermediate ancestral population between the pair and Population 0 minimises $\xi$ in (11). (c) Repeating step (b), now $1D$ is the optimal pair of populations in $\{1, B, D\}$ (the children of Population 0). After $K - 2 = 2$ steps, the resulting tree is binary and the algorithm stops.

parameters to estimate, allowing the trees to be compared using the values of $\xi$ in (11). A global search over all possible trees is infeasible even for moderate $K$. Instead, we first use a pairwise clustering strategy, as illustrated for $K = 4$ sampled populations in Fig 3. Starting with the independent-descent model, at each step an intermediate ancestral population is added between Population 0 and two of its child populations, chosen to minimise $\xi$.

After $K - 2$ steps a binary tree is obtained which we then seek to improve. For each tip node $k$, chosen in random order, we consider each branch in the current tree as an alternative location for the parent of $k$, fitting each of these $2(K - 1)$ trees and choosing the one that minimises $\xi$, which may be the current tree in which case no change occurs.

In the clustering phase there are $K - 2$ merge steps, and at the $j$th step there are $\binom{K - j}{2}$ pairs of populations to consider merging. Overall we require $\mathcal{O}(K^3)$ solutions of the non-negative least-square optimization problem (11), for which we use the Lawson-Hanson algorithm [16]. The improvement phase of the algorithm scales with $K^2$, because there are $K$ tips to consider relocating, and $2(K - 1)$ locations to consider for each of them. In practice each step in the clustering phase can be solved easily using a warm-start strategy for initialization: each new fitting can be initialized using the tree and parameters inferred in the previous step. Consequently the computational burden of the improvement phase is usually higher. Solving (11) also requires computation of the $S_{kk'}$, which is linear in $m$, the number of SNVs. This computation only has to be performed once, after which there is no further dependence on $m$, making the procedure feasible for any number of SNVs. See S2 Text for more details.

## Simulation study design

We simulated the allele frequency at each SNV in populations $A$ to $E$ that evolved with constant, large size according to the tree of Fig 4. Sites were simulated independently (no LD). We first simulated the allele frequency $p$ in the unobserved ancestral population (Population 0) from a beta(0.4,0.4) distribution, so that 19% of SNVs have $p \notin (0.01, 0.99)$ and 36% have $p \notin (0.05, 0.95)$. The allele frequency in each other population was simulated from a beta distribution with moments (8). Finally, we randomly sampled gametes from a binomial distribution using the simulated value of the allele frequency in each of populations $A$ to $E$.

The top two rows of Table 2 show simulation parameters, ordered from the most informative scenario (S1) to the least informative (S6) in terms of the number of correct tree inferences (Table 2, final row). While all population allele frequencies remain positive under our model,

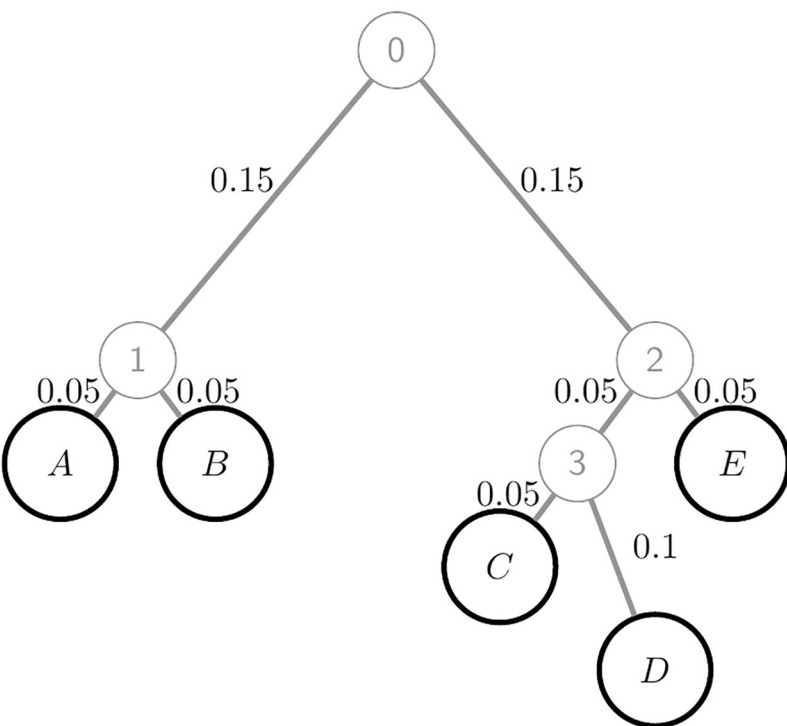

**Fig 4. Population tree used for the simulation study.** Each branch length $\theta_q$ is shown next to the corresponding branch. In the simulation with admixture, the parent of Population $C$ has a contribution $\alpha$ from Population 1.

genetic drift between Population 0 and the sampled populations increases the proportions of low-MAF SNVs, and many sites are monomorphic in the sample (Table 2, third row).

For each simulated dataset we used (7) to compute the pairwise estimator $\hat{F}_{ST}^{H}(kk')$ for $k, k' \in \{A, B, C, D, E\}$. Then, treating the population tree as unknown, we jointly inferred it and $\theta$ for each branch, and computed our novel estimators $\tilde{F}_{ST}^{W}(kk')$ and $\tilde{F}_{ST}^{H}(kk')$.

We repeated the S4 simulations but now one of the sampled populations is an admixture of two highly-diverged ancestral populations: Population C descends from a parent population with fraction $\alpha$ of its alleles drawn from Population 1 and the remaining fraction $1 - \alpha$ drawn from Population 3. In S4 Text we derive the exact $F_{ST}^{H}$ values under this model.

## The 1000 Genomes dataset

We applied our joint estimation of tree and $\theta$ values to data from Phase 3 of the 1000 Genomes Project [17, 18], from 2 504 individuals sampled in 26 populations classified into five continental-scale "superpopulations" (Table 3). We included all available diallelic SNPs across the 22

**Table 2. Details of six simulation scenarios based on the population tree of Fig 4.** # denotes "number of". $10^4$ simulation replicates were performed for each scenario. An SNV has at least one copy of a minor allele in at least one population, there is no MAF threshold.

| Parameter | S1 | S2 | S3 | S4 | S5 | S6 |
|---|---|---|---|---|---|---|
| # gametes sampled per pop | 25 | 10 | 2 | 25 | 10 | 10 |
| # sites simulated | $10^4$ | $10^4$ | $10^4$ | $10^3$ | $10^3$ | $10^2$ |
| % of sites that are SNVs | 69.72 | 65.95 | 52.42 | 69.7 | 66.0 | 66 |
| % of trees correctly inferred | 100 | 100 | 99.99 | 99.98 | 99.57 | 70.31 |

**Table 3. Description of 1000 Genomes Project data, and $F_{ST}^W$ estimates.** Pop = population. The superpopulation label is used for discussion but not in any analysis. $F_{ST}^W$ measures divergence of the population from the inferred global ancestral population.

| Pop code | Superpop code | Ancestry descriptor; place sampled | Sample size (gametes) | $\tilde{F}_{ST}^W$ |
|---|---|---|---|---|
| ACB | AFR | African Caribbeans; Barbados | 192 | 0.028 |
| ASW | AFR | African Americans; SW USA | 122 | 0.054 |
| ESN | AFR | Esan; Nigeria | 198 | 0.016 |
| GWD | AFR | Gambian; Western Divisions, Gambia | 226 | 0.017 |
| LWK | AFR | Luhya; Webuye, Kenya | 198 | 0.014 |
| MSL | AFR | Mende; Sierra Leone | 170 | 0.015 |
| YRI | AFR | Yoruba; Ibadan, Nigeria | 216 | 0.017 |
| CLM | AMR | Colombians; Medellin, Colombia | 188 | 0.208 |
| MXL | AMR | Mexicans; Los Angeles | 128 | 0.242 |
| PEL | AMR | Peruvians; Lima, Peru | 170 | 0.303 |
| PUR | AMR | Puerto Ricans; Puerto Rico | 208 | 0.191 |
| CDX | EAS | Chinese Dai; Xishuangbanna, China | 186 | 0.307 |
| CHB | EAS | Han Chinese; Bejing | 206 | 0.305 |
| CHS | EAS | Southern Han Chinese | 210 | 0.308 |
| JPT | EAS | Japanese; Tokyo | 208 | 0.307 |
| KHV | EAS | Kinh; Ho Chi Minh City, Vietnam | 198 | 0.300 |
| CEU | EUR | NW Europeans; Utah | 198 | 0.257 |
| FIN | EUR | Finnish; Finland | 198 | 0.263 |
| GBR | EUR | British; England and Scotland | 182 | 0.259 |
| IBS | EUR | Iberians; Spain | 214 | 0.249 |
| TSI | EUR | Toscani; Italia | 214 | 0.251 |
| BEB | SAS | Bengali; Bangladesh | 172 | 0.231 |
| GIH | SAS | Gujarati Indian; Houston, Texas | 206 | 0.237 |
| ITU | SAS | Indian Telugu; UK | 204 | 0.234 |
| PJL | SAS | Punjabi; Lahore, Pakistan | 192 | 0.230 |
| STU | SAS | Sri Lankan Tamil; UK | 204 | 0.233 |

autosomes, totalling 72M, with a binary coding indicating presence/absence of the major allele. The four AMR populations are strongly affected by historical admixture, including from different Native American source populations who are closest to the EAS superpopulation among the study populations. Estimated fractions of Native American ancestry are PEL 0.77, MXL 0.47, CLM 0.26 and PUR 0.13 [19]. The remaining ancestry comes mainly from European populations best represented among our study populations by IBS, but nearly 10% of the ancestry of AMR individuals is African (both European and African ancestry fractions are highest in PUR and lowest in PEL). ASW and ACB individuals also show some European admixture, but their ancestry is predominantly African (estimated fractions ACB 0.88, ASW 0.76 [19]). Some ASW individuals also show substantial Native American ancestry.

As well as the population-level analysis of all 2 504 individuals, we performed individual-level analyses for a subsample of five individuals from each of six populations: three AMR populations (CLM, MXL, PUR) and one population from each of AFR, EAS and EUR, namely the MSL, CHB and IBS populations. See S1 Table for identifiers of the selected individuals. Of the 72M SNVs in the full dataset, 13.4M remained SNVs in the 30-individual dataset. Of these 4.7M and 1.5M had one and two copies of the minor allele, respectively, while 1.7M had over 20 copies of the minor allele. We also performed principal component (PC) analysis which is a standard approach to visualising individuals based on their genome-wide genotypes. However,

we did not apply the usual standardising of the SNV variables. Due to the absence of an MAF threshold, with standardisation the first five PCs are dominated by the 4.7M singleton sites and only differentiate the five MSL individuals from each other and the rest of the sample.

Computation for the 26-population and 30-individual analyses each took around 10 minutes on a standard desktop computer. The different numbers of SNVs (72M and 13.4M) has little impact on computing time, and the first (clustering) phase of the tree-inference algorithm required just a few seconds for both analyses, with the improvement phase requiring most of the computing time.

## Results

### Generalisation of $F_{ST}^W$ and $F_{ST}^H$ to tree-structured populations

Underlying all our results is the parameter $\theta_k \in [0, 1]$, which equals $F_{ST}^W(k)$ in the independent-descent model of Fig 1. See under "Statistical model and definitions of $F_{ST}$ in the classical setting" in **Materials and Methods** for a review. Given a pre-specified tree topology (see for example Fig 2) and assuming (8), we define $F_{ST}^W$ and $F_{ST}^H$ and express them in terms of the $\theta_k$ (see S1 Text for proofs).

**Theorem 1**.

$$F_{ST}^W(kk') = \frac{1}{p(1-p)}\mathbb{V}\mathrm{ar}[p_M|p] = 1 \quad - \prod_{q \in \mathcal{Q}(kk')}(1 - \theta_q) \tag{12}$$

$$\begin{aligned}
F_{ST}^H(kk') &= \frac{1}{2}\{\mathbb{C}\mathrm{or}[x_k, y_k|p_M] + \mathbb{C}\mathrm{or}[x_{k'}, y_{k'}|p_M]\} \\
&= 1 - \frac{1}{2}\left\{\prod_{q \in \mathcal{R}(kk')}(1 - \theta_q) + \prod_{q \in \mathcal{R}(k'k)}(1 - \theta_q)\right\}
\end{aligned} \tag{13}$$

*where a product over an empty set is defined to equal one. See* Table 1 *for definitions of M = M (kk'), $\mathcal{Q}$ and $\mathcal{R}$.*

From (12) and (13) we see that $F_{ST}^W(kk')$ and $F_{ST}^H(kk')$ are functions of disjoint sets of $\theta$ coefficients. Fig 2 illustrates the $\theta$ coefficients that contribute to $F_{ST}^W$ and $F_{ST}^H$. In particular, $F_{ST}^W(kk')$ measures the shared genetic variation of populations $k$ and $k'$ relative to Population 0, and so depends on $\theta$ values for tree branches between populations 0 and $M$. If $M = 0$, then $F_{ST}^W(kk') = 0$ while if $k = k'$ then $F_{ST}^W(k)$ measures the divergence of Population $k$ from Population 0. Both $F_{ST}^W(kk')$ and $F_{ST}^H(kk')$ are invariant to switching $k$ and $k'$. The value of $F_{ST}^W(kk')$, but not $F_{ST}^H(kk')$, can change if new populations are included or existing populations (other than $k$ and $k'$) are removed such that the ancestral population changes.

We compute our novel estimators $\tilde{F}_{ST}^W(kk')$ and $\tilde{F}_{ST}^H(kk')$ by replacing each $\theta_q$ with $\hat{\theta}_q = 1 - \exp(\hat{\beta}_q)$ in (12) and (13), respectively, where the $\hat{\beta}_q$ are obtained from the optimisation (11) in **Material and Methods**. While our approach explicitly allows for linkage disequilibrium (LD) due to population structure, LD due to tight linkage is not modelled. Any effects of linkage on estimates are expected to be small for random samples from large, outbred species.

### Simulation study

All the estimators considered here have low bias and so RMSE (Table 4) is close to the standard error. We typically have $\mathrm{RMSE}(\tilde{F}_{ST}^W(kk')) > \mathrm{RMSE}(\tilde{F}_{ST}^H(kk'))$ when the true values are similar (Table 5), the lower precision of $\tilde{F}_{ST}^W$ reflecting that it estimates lengths of branches close to the

**Table 4. $F_{ST}^H(kk')$ and the RMSE of $\tilde{F}_{ST}^H(kk')$.** Based on $10^4$ replicates of each simulation scenario. In brackets is the ratio of the RMSE of $\hat{F}_{ST}^H(kk')$, the pairwise estimator of [7], to that of the novel tree-based estimator $\tilde{F}_{ST}^H(kk')$; values >1 indicate that $\tilde{F}_{ST}^H(kk')$ performs better than $\hat{F}_{ST}^H(kk')$. The final row gives the average of the RMSE ratios over the 10 population pairs.

| $kk'$ | $F_{ST}^H(kk')$ | $\mathrm{RMSE}(\tilde{F}_{ST}^H) \times 10^4$ $(\mathrm{RMSE}(\hat{F}_{ST}^H)/\mathrm{RMSE}(\tilde{F}_{ST}^H))$ | | | | | |
|---|---|---|---|---|---|---|---|
| | | S1 | S2 | S3 | S4 | S5 | S6 |
| AB | 0.050 | 17 (1.00) | 29 (1.00) | 123 (1.00) | 54 (1.00) | 90 (1.00) | 264 (1.08) |
| CE | 0.074 | 23 (0.92) | 33 (0.99) | 118 (1.07) | 72 (0.92) | 102 (0.99) | 305 (1.06) |
| CD | 0.075 | 22 (1.00) | 34 (1.00) | 129 (1.00) | 70 (1.00) | 107 (1.00) | 331 (1.01) |
| DE | 0.097 | 24 (1.03) | 34 (1.04) | 117 (1.06) | 76 (1.03) | 108 (1.05) | 333 (1.08) |
| AE,BE | 0.192 | 34 (1.01) | 42 (1.04) | 108 (1.08) | 109 (1.01) | 133 (1.04) | 416 (1.05) |
| AC,BC | 0.213 | 35 (1.07) | 42 (1.08) | 106 (1.10) | 110 (1.06) | 130 (1.09) | 418 (1.09) |
| AD,BD | 0.233 | 36 (1.10) | 43 (1.10) | 106 (1.10) | 113 (1.10) | 134 (1.11) | 426 (1.10) |
| Mean RMSE ratio | | (1.03) | (1.05) | (1.07) | (1.03) | (1.05) | (1.07) |

root of the population tree, whereas $\tilde{F}_{ST}^H$ relates to branches near the tips. While $F_{ST}^H(k) = 0$ by definition, $F_{ST}^W(k)$ measures divergence for a single population or individual.

We observe $\mathrm{RMSE}(\tilde{F}_{ST}^H) < \mathrm{RMSE}(\hat{F}_{ST}^H)$ for 38 of the 42 values reported in Table 4. The mean RMSE ratio over the 10 population pairs ranges from 1.03 to 1.07. This small gain in efficiency of $\tilde{F}_{ST}^H$ over $\hat{F}_{ST}^H$ is consistent across simulation scenarios and comes with the important gains in visualisation and interpretability of our approach. $\hat{F}_{ST}^H(AB)$ performs almost as well as $\tilde{F}_{ST}^H(AB)$, reflecting that there are no populations sufficiently close to $A$ and $B$ to provide useful information. Conversely, for all the population pairs only connected in the population tree via the root, $\tilde{F}_{ST}^H$ is superior to $\hat{F}_{ST}^H$. For example, in estimating $F_{ST}^H(AC)$, allele frequencies in population $B$ are informative about frequencies in the path between $A$ and $C$, and only $\tilde{F}_{ST}^H$ exploits this information. The exception is the pair $CE$, for which the allele frequencies in $D$ convey some relevant information, but it does not always improve inferences of $F_{ST}^H(CE)$, reflecting that $D$ is highly diverged from the path connecting $C$ with $E$.

In S3, the five sampled populations are each represented by a sample of size two gametes, and so $F_{ST}^W$ and $F_{ST}^H$ describe the coancestries among five individuals. For S6 with only 66 polymorphic SNVs, the population tree was correctly inferred in only 70% of simulations (Table 2, final row), but enough correct features of the tree were extracted to improve inference such

**Table 5. $F_{ST}^W(kk')$ and the RMSE of $\tilde{F}_{ST}^W(kk')$ in three simulation scenarios.** Based on $10^4$ replicates in each simulation scenario (see Table 2).

| $kk'$ | $F_{ST}^W(kk')$ | $\mathrm{RMSE}(\tilde{F}_{ST}^W) \times 10^4$ | | |
|---|---|---|---|---|
| | | S1 | S3 | S6 |
| AB | 0.150 | 53 | 100 | 578 |
| CE | 0.150 | 52 | 91 | 588 |
| DE | 0.150 | 52 | 91 | 594 |
| A,B | 0.192 | 55 | 153 | 631 |
| E | 0.192 | 55 | 154 | 636 |
| CD | 0.192 | 56 | 105 | 636 |
| C | 0.233 | 56 | 150 | 649 |
| D | 0.273 | 58 | 150 | 665 |

**Table 6. Performance of estimators under admixture of highly-diverged populations.** RMSE values are averages over 10 pairs of observed populations, and the correlations (Corr) are over these 10 estimate-parameter pairs. $10^4$ simulation replicates were performed under the model of Fig 4 except that a fraction $\alpha$ of the alleles in the parent of Population $C$ now come from Population 1, with the remaining alleles continuing to come from Population 3. The admixing populations are highly diverged: $F_{ST}^H(1,3) = 0.175$.

| | Admixture proportion $\alpha$ | | |
|---|---|---|---|
| | **0.01** | **0.05** | **0.1** |
| RMSE($\hat{F}_{ST}^H$)/RMSE($\tilde{F}_{ST}^H$) | 1.03 | 0.98 | 0.90 |
| Corr($\hat{F}_{ST}^H$,$F_{ST}^H$) | 0.996 | 0.995 | 0.995 |
| Corr($\tilde{F}_{ST}^H$,$F_{ST}^H$) | 0.996 | 0.995 | 0.993 |

that $\tilde{F}_{ST}^H$ showed the largest relative improvement over $\hat{F}_{ST}^H$ in this low-information scenario (Table 4).

When the parent of $C$ is constructed as an admixture of Populations 1 and 3 (see Materials and methods), the small advantage of $\tilde{F}_{ST}^H$ over $\hat{F}_{ST}^H$ is retained only when the admixture proportion is low, due to the high divergence between the two parent populations. However the correlation between estimate and true value remains very high for both estimators (Table 6). We show in the example below that the visualisation and interpretation advantages of our tree-based approach are evident in the presence of substantial admixture.

## 1000 Genomes population analysis (26 populations, $n = 2504$)

The single-population $\tilde{F}_{ST}^W$ values, measuring the divergence of each of the 26 populations from the inferred global ancestral population (Table 3), are lowest for the AFR populations (0.01—0.05) and highest for PUR and the EAS populations (0.30—0.31). Greater divergence of non-AFR populations may reflect an out-of-Africa bottleneck. The AMR superpopulation has the greatest range of $\tilde{F}_{ST}^W$, 0.19—0.30, with values ordered by the level of African admixture. The average of the four values is 0.24, close to the average individual-specific $F_{ST}$ of 0.23 for AMR reported by [20].

The $\tilde{F}_{ST}^H$ values, measuring divergence between pairs of populations, show a familiar pattern for human population genetic studies, with the largest values comparing AFR with non-AFR populations, particularly the 35 AFR-EAS population pairs (Fig 5A). The largest value is $\tilde{F}_{ST}^H$(CHS,ESN) = 0.165. Within superpopulations, the maximum $\tilde{F}_{ST}^H$ values are 0.007 for SAS, 0.011 for EUR, 0.013 for EAS, 0.031 for AFR and 0.068 for AMR.

The inferred population tree (Fig 5B) reveals more structure than is evident from the matrix of $\tilde{F}_{ST}^H$ values. As expected, the AFR, EAS, EUR and SAS superpopulations each cluster together, but now we can also see that the admixed AFR populations, ACB and ASW, are closer to non-AFR populations, and IBS is the EUR population closest to non-EUR populations, reflecting the contribution of Iberians to AMR populations. The longest branch in the tree, which connects ASW with PUR, lies on the path between every AFR and non-AFR population pair, giving PUR a central position among the 1000 Genomes populations. Consequently, $\tilde{F}_{ST}^H(kk')$ where $k \in$ AFR and $k' \notin$ AFR is well approximated by $\tilde{F}_{ST}^H(k\text{PUR}) + \tilde{F}_{ST}^H(\text{PUR}k')$.

The root of the inferred tree separates West African (ESN, GWD, MSL, YRI) from all other populations, consistent with the origins of modern humans in Africa. The largest genetic distances are between West-African and EAS populations, which reflects their geographical separation and low historical migration.

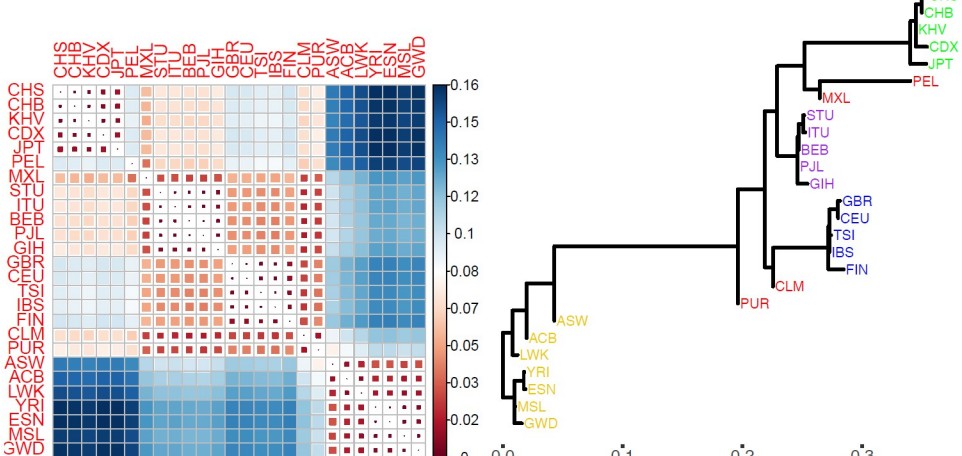

**Fig 5. Tree-based inferences from the 26 populations of the 1000 Genomes dataset. A**: $\tilde{F}_{ST}^H$ values (see scale for colour coding) for each pair of populations. **B**: The inferred population tree, with horizontal branch lengths corresponding to coancestry parameter estimates $\hat{\theta}$ (see $x$-axis scale). Vertical distances have no meaning and are for display purposes only.

Fig 5B gives a visual representation of actual genetic variation among the 1000 Genomes populations. The observed variation is strongly influenced by historical processes of splitting and divergence, but the tree may not accurately reflect actual historical events. For example, the inferred global ancestral population provides a convenient reference for describing components of genetic variance, but may not accurately represent any actual population in human history.

While admixture events are not explicitly modelled, effects of admixture can be discerned in the current patterns of genetic similarity. The AMR populations are divergent from each other and other populations, with CLM closest to EUR and PEL and MXL closest to EAS populations, corresponding with their levels of admixture outlined above. PEL is the population most distant from its nearest neighbour, MXL, with $\tilde{F}_{ST}^H$(MXL,PEL) = 0.038.

The high correlation of $\tilde{F}_{ST}^H$ and $\hat{F}_{ST}^H$ values (0.984) is driven by the similarity of the two estimators for the largest genetic distances (Fig 6A), whereas there are substantial differences between them over most of the range. The comparisons between PUR and the five EUR populations give the largest values of $\tilde{F}_{ST}^H - \hat{F}_{ST}^H$, between 0.027 and 0.030 (Fig 6B). There are three comparisons for which $\tilde{F}_{ST}^H - \hat{F}_{ST}^H < -0.014$, each involving a EUR and an EAS population (TSI-CDX, KHV-IBS, and CDX-IBS).

## 1000 Genomes individual analysis (6 populations, $n = 30$)

There is good agreement in $\tilde{F}_{ST}^W$ values between the individual and population analyses (Table 7), despite the great difference in sample size and populations sampled. This is important because $\tilde{F}_{ST}^W$ is based on the inferred ancestral population, but estimates can still be comparable across very different datasets.

Fig 7 shows a PC plot and the inferred tree for the 30 individuals. The two plots convey similar information, with the tree giving finer detail about shared and non-shared components of variance among the individuals plus interpretability from horizontal branch lengths corresponding to $\theta$ and hence $F_{ST}^W$ and $F_{ST}^H$ estimates. The CLM, MXL and PUR population labels

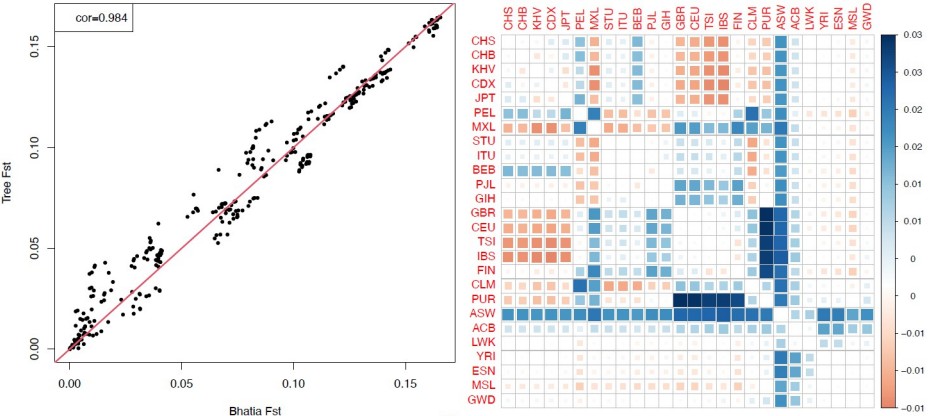

**Fig 6. A**: pairwise estimator $\hat{F}_{ST}^H$ [7] (*x* axis) and tree-based estimator $\tilde{F}_{ST}^H$ (*y* axis) for all 325 pairs of populations in the 1000 Genomes dataset. **B**: colour-coded values of $\tilde{F}_{ST}^H - \hat{F}_{ST}^H$, which is largest for comparisons of PUR with all 5 EUR populations (dark-blue squares).

indicate location of sampling, but they do not accurately reflect genetic structure because of the high within-group diversity, with many instances of between-group pairs of individuals being genetically closer to each other than within-group pairs. One MXL individual is genetically closer to all of the IBS sample than to any other MXL individual.

Table 7 shows a wide range of between-individual divergence ($\tilde{F}_{ST}^H$) in the admixed AMR populations. The non-AMR populations are more homogeneous, although there is higher between-pair divergence in IBS than in CHB or MSL, which may reflect some migration from the Americas. S1 Fig shows the corresponding tree when the non-AMR individuals are pooled into 3 population samples. This analysis reduces computing time with little loss of information.

## Discussion

We have extended the definitions of the genetic distance $F_{ST}$ to the tree-structured multi-population setting, showing that correlation and mismatch probability definitions of $F_{ST}$ measure shared and non-shared genetic variation.

Compared to the independent descent population model, the tree structure describes covariances of allele frequencies across actual populations. Previous authors have allowed for

**Table 7. For 5 individuals from the population in column 1, column 2 gives the range of $\tilde{F}_{ST}^W(k)$ measuring divergence from the inferred ancestral population, and (in brackets) the corresponding population-level value from Table 3**. Also shown are the ranges over the 10 within-population pairs of individuals of $\tilde{F}_{ST}^W$ and $\tilde{F}_{ST}^H$, measuring respectively shared genetic variance and between-pair divergence.

| | Individual | Within-pop pairs | |
|---|---|---|---|
| **Pop** | $\tilde{F}_{ST}^W(k)$ | $\tilde{F}_{ST}^W(kk')$ | $\tilde{F}_{ST}^H(kk')$ |
| MSL | 0.009 − 0.013 (0.015) | 0.009 − 0.012 | 0.001 − 0.004 |
| CHB | 0.31 − 0.32 (0.31) | 0.31 − 0.31 | 0.006 − 0.008 |
| IBS | 0.25 − 0.26 (0.25) | 0.25 − 0.26 | 0.004 − 0.015 |
| CLM | 0.21 − 0.26 (0.21) | 0.21 − 0.24 | 0.004 − 0.030 |
| PUR | 0.19 − 0.24 (0.19) | 0.19 − 0.23 | 0.002 − 0.032 |
| MXL | 0.23 − 0.27 (0.24) | 0.21 − 0.23 | 0.001 − 0.062 |

non-zero covariances [6, 8, 9, 11] without specification of a correlation structure. While the tree structure may not fully reflect the evolutionary history of the population studied, adopting it brings two attractive features. Firstly, from a theoretical perspective it leads to closed-form expressions for both $F_{ST}^W$ and $F_{ST}^H$. In contrast, for unstructured covariances [8] note that "it is not possible to find estimates for each [allele correlation parameter, corresponding to our $F_{ST}^W$] . . .when sampled populations have correlated sample allele frequencies", although they go on to point out that an approximate ranking is available. In our framework the number of $\theta$ parameters (= $2(K-1)$) is less than the number of pairs of populations ($K(K-1)/2$) and hence less than the number of empirical moments available for inference, allowing estimation of both $F_{ST}^W$ and $F_{ST}^H$. Secondly, from a practical point of view, our framework allows the joint inference of tree structure and both $F_{ST}$ parameters, for which we have developed an efficient procedure, and comes with a natural representation of the divergence between populations. Although methods for inferring a population tree from allele frequency data are already available, including `Treemix` [21] and Neighbor Joining [22], our procedure is unique in performing joint inference of $F_{ST}$ and the tree, which allows sharing of information about allele frequencies in ancestral populations and a range of options for visualising genetic structure by combining homogeneous groups of individuals.

The inferred tree can reflect historical population splitting and divergence, but it may not accurately reconstruct the evolutionary history of the populations studied because there is no explicit role for admixture. Instead, the tree provides a visual representation of the actual genetic variation across the populations via an approximately best-fitting tree, with branch lengths that are interpretable as $F_{ST}$ values. However, in the presence of admixture it may be the case that no tree can accurately capture the genetic structure. In S4 Text we have taken a first step towards showing that our tree-based procedure generalises naturally to admixture graphs, by illustrating in a particular case the generalisations of the definitions of $F_{ST}^H$ and $F_{ST}^W$, and the computational formulas in Theorem 1. Generalising our logarithmic least-squares inference procedure (11) to admixture graphs will be explored in future work.

Our methods also provide a novel approach to describing coancestry among sets of diploid individuals, treating each as a population of two gametes. This is not practical for pairwise estimators of $F_{ST}$ because of inadequate information about reference allele frequencies, whereas other individuals and populations inform about them in our approach. Any pair of individuals are related through many ancestral lineages of varying lengths. Pedigree-relatedness captures only very short lineage paths (within the known pedigree), whereas $\tilde{F}_{ST}^H$ for two individuals is affected by all lineage paths connecting them, which can be useful to construct adjustments for even subtle population structure in heritability analyses and genetic association analyses. Currently we do not model LD among the markers and so cannot accommodate closely related individuals, but close relatives are also usually excluded from GWAS.

Since the seminal contribution of Lewontin [23], there has been interest in comparing genetic diversity within and between human populations. For example, within-Africa genetic differences were reported to be larger than differences between Eurasians and Africans [24]. Estimates $\tilde{F}_{ST}^W$ provide a convenient way to quantify such comparisons. Fig 7B shows that, for these six populations, diversity is much lower within CHB and MSL than for any between-population comparison, but for CLM, MXL and PUR within-population and between-population $\tilde{F}_{ST}^W$ can be of similar magnitude.

In S3 Text, our inference procedure is extended to distinguish the effects of inbreeding from other evolutionary forces, based on comparing expected and observed heterozygosities to estimate individual inbreeding coefficients jointly with coancestry coefficients measuring the effects of other evolutionary processes. Another extension is to model the component of

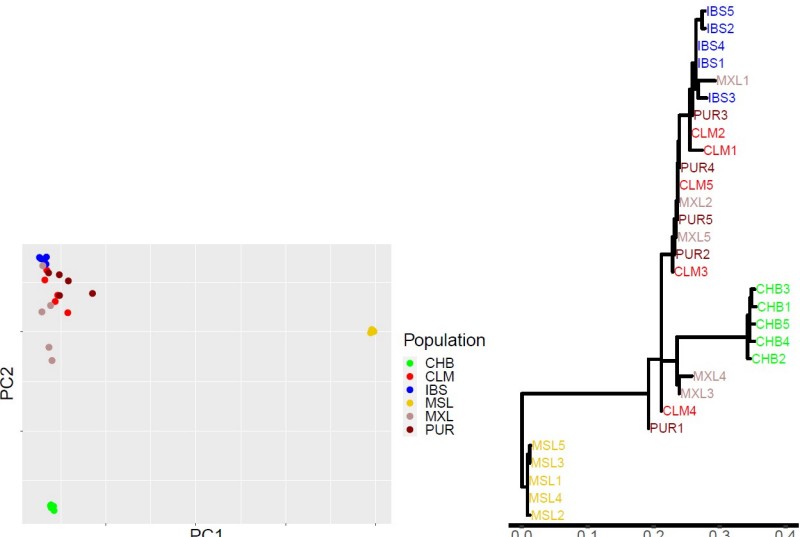

**Fig 7. A**: First two principal components (explaining 29% of variance) from 13.4M unstandardised SNVs in a sample of 30 individuals from the 1000 Genomes dataset (5 each from six populations as indicated in the legend box). **B**: Inferred tree for the 30 individuals, with horizontal branch lengths corresponding to coancestry parameter estimates $\hat{\theta}$.

variance shared by a set of populations or individuals, rather than just pairs, which requires replacing $\mathcal{Q}$ in (12) with $\mathcal{P}(k_1) \cap \mathcal{P}(k_2) \cap ... \cap \mathcal{P}(k_p)$.

Locus-specific $F_{ST}$ values that diverge from genome-wide averages have long been used to help identify the effects of natural selection [25–28], with some methods depending explicitly on a population tree [29, 30]. Our tree-based approach may be able to increase the power of such methods, and simultaneous inference of all $\theta$ parameters should lead to better characterisation of the selection effect, which will also be explored in future research.

Values of $\tilde{F}_{ST}^H$ for pairs of individuals, illustrated graphically in Fig 7B, can be useful in assessing forensic match probabilities comparing alleged and alternative sources of a crime-related DNA sample [31]. For practical reasons $F_{ST}$ has been estimated at the level of populations [32, 33] but the relevant value of $F_{ST}$ measures relatedness of pairs of individuals, the alleged and alternative sources of a crime-related DNA sample. While it is not typically possible to estimate $F_{ST}$ for all possible alternative sources, a range of $F_{ST}$ values over many pairs of individuals can indicate values that may be relevant to a particular case. We can also include population data from a forensic database, which can be used to ensure a representative reference population that is the same over different cases. Forensic DNA profiling primarily uses short tandem repeat loci rather than SNVs, and these have different $F_{ST}^H$ values due to a different mutation process, but SNV-based DNA profiling is becoming more common [34].

## Supporting information

**S1 Text. Proofs.**
(PDF)

**S2 Text. Details of the tree-based inference algorithm.**
(PDF)

**S3 Text. Joint estimation of inbreeding and coancestry effects.**
(PDF)

**S4 Text. $F_{ST}^{H}$ for the simulation model with admixture.**
(PDF)

**S1 Fig. Inferred tree showing coancestry among the 15 individuals and 3 population samples.**
(PDF)

**S1 Table. 1000 Genomes Project identifiers.**
(PDF)

## Acknowledgments

The authors gratefully acknowledge Dr Angad Johar for assistance with extracting the 30-individuals subset of the 1000 Genomes Project data.

## Author Contributions

**Formal analysis:** Tristan Mary-Huard, David Balding.

**Funding acquisition:** Tristan Mary-Huard, David Balding.

**Methodology:** Tristan Mary-Huard, David Balding.

**Software:** Tristan Mary-Huard.

**Writing – original draft:** Tristan Mary-Huard, David Balding.

**Writing – review & editing:** Tristan Mary-Huard, David Balding.

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
