## [Decision Letter · Decision Letter 0]

8 Mar 2022

Dear Dr Mary-Huard,

Thank you very much for submitting your Research Article entitled 'Fast and accurate joint inference of coancestry parameters for populations and/or individuals' to PLOS Genetics.

The manuscript was fully evaluated at the editorial level and by independent peer reviewers. The reviewers appreciated the attention to an important problem, but raised some substantial concerns about the current manuscript. Based on the reviews, we will not be able to accept this version of the manuscript, but we would be willing to review a much-revised version. We cannot, of course, promise publication at that time.

If you decide to revise the manuscript for further consideration at PLOS Genetics, please aim to resubmit within the next 60 days, unless it will take extra time to address the concerns of the reviewers, in which case we would appreciate an expected resubmission date by email to plosgenetics@plos.org.

[LINK]

We are sorry that we cannot be more positive about your manuscript at this stage. Please do not hesitate to contact us if you have any concerns or questions.

Yours sincerely,

B.S Weir

Guest Editor

PLOS Genetics

Scott Williams

Section Editor: Human Variation

PLOS Genetics

Reviewer's Responses to Questions

**Comments to the Authors:**

Reviewer #1: I think this paper is very interesting and ultimately may be very useful. However, I do have two major areas of concern.

The first is presentational. There is broad interest among empirical biologists in estimating FST, but this paper is written for a strictly mathematical audience. A great deal more care needs to be taken in terms of clarity of mathematical definitions and more standard use of terms. I’ll try to flag some of these issues on a case by case basis below, but overall the paper could be a lot friendlier to a non-mathematical reader (ultimately, the lion’s share of the potential readership).

The second concern I have is more crucial. The method outlined here is developed in the context of populations without migration, leading to an unambiguous and unreticulated branching tree of population relationships. This situation strictly applies to a very, very small subset of applications of Fs, excluding even the human history to which the method is applied to here (as indicated in line 196…). Before this is published, I would very much like to see it applied to simulations of populations that experience migration, so that the approach can be ground-truthed before it goes into mass use. The core assumption of the method is at such odds with most biological patterns in the relevant use cases that it is crucial to determine whether the method gets sensible answers in this more realistic context.

Specific comments:

28: It is more usual to refer to “tips” rather than “leaves” – whatever choice you make be sure to define the term.

48-52: This notion is central to much of the discussion of the paper. It is worth a bit more unpacking to make clear what you mean by shared and unshared, and how these relate to the different FST’s. Also, it is more challenging for the reader at this point when the only figure so far is figure 1, where the shared and unshared distinctions don’t matter.

67-68: The standard term in population genetic is allele frequency not allele fraction. I’d suggest changing the term used, but if not define allele fraction.

67-68: Lower case f has many other meanings in the literature of structured populations (e.g. probability of identity by descent), whereas p is used for allele frequency in the vast majority of papers. Please, please don’t use f for this quantity; p is a good alternative.

71: Define divergence, especially in terms of theta. A key parameter here ( theta) is essentially undefined.

Table 1: Many readers will not know the \\ notation – spell out the intent here. Discuss the meaning of Q, P and R in the text

104-106: I didn’t follow this paragraph

Eq 11: This equation is crucial, yet no justification or explanation is provided. Explain what this equation is doing, and justify this choice for beta hat.

147: It seems out of place to have this line about FST_H when the topic of the preceding is FST_W. Is this a typo, or else can more context be given?

177: Define MAF

Simulation section (174-190): This section seems wholly inadequate to describe to the reader how they may recreate the procedure followed here.

188: Says “See Methods”, but this is part of the Methods section. Where is this intended to point?

218: Giving the run time is not meaningful with info about the computer infrastructure used.

Table 4: Perhaps give bias of FST_W hat for comparison?

Table 5: My main impression from Table 5 is that these RMSE ratios are all very close to 1, and the effect of the new procedure is small.

Results, somewhere: It would be very useful to know how dependent the method is on the choice of populations to be included in the sampling and analysis. I imagine that if populations that are more closely related than average are included in the samples that one could get quite different answers than if the samples are more independently taken. Give that this non-random choice of populations is the norm in empirical work, it is crucial to know how this affects the method. (I can see it likely that this method may improve on alternatives in this regard, but it would be nice to see the effects.)

Reviewer #2: There is a plethora of Fst estimators that differ in the particular definition being assumed and/or the estimation method being used. This has generated much confusion as it is difficult or even impossible to compare estimates of population differentiation among different studies whether they deal with the same or different species. Any study attempting to clarify the relationship between different Fst estimators is therefore very much welcome.

The most insightful contribution of this study is the clarification of the relationship between the correlation-based estimator of Weir and colleagues (Weir and Cockerham 1984; Weir and Hill 2002) and the IBD formulation of Hudson et al. (1992). The authors show that when applied to pairs of populations, and under a scenario of successive population fission events, these estimators provide complementary information about population structure, namely the lengths of shared and non-shared branches between the focal populations and the ancestral population. Based on this insight, the authors’ also propose new tree-based estimators of Fst corresponding to those of Weir and Hudson.

Overall, I don’t have any major concerns. The main observation I would make is that, as opposed to the formulation of Weir and Goudet (2017), the authors’ formulation does not consider inbreeding. In principle, this seems to be done for the sake of simplicity but I would have very much liked to see a discussion of the extent to which the authors’ framework relates to the much more general formulation of Weir and Goudet, which is only briefly mentioned in the manuscript.

Some minor issues relate to the fact that some of the properties of the new estimators are duly mentioned in the material and methods but are not revisited in the discussion. This is important because many or even most potential users of the new estimators are unlikely to read the M&M. For example, the authors acknowledge that both new estimators of pairwise Fst can be affected by the addition or removal of populations. This is a very important observation that users of the method should keep in mind when interpreting their results.

Another comment relates to the fact that when evaluating the performance of the new population-specific estimator of Fst, the authors only present results for three of the five scenarios (S1, S3, and S6). It is unclear to me why this is the case and I think the authors should clarify this discrepancy between the approach used to evaluate the two new estimators of Fst.

**Have all data underlying the figures and results presented in the manuscript been provided?**

Reviewer #1: Yes

Reviewer #2: Yes

PLOS authors have the option to publish the peer review history of their article (what does this mean?). If published, this will include your full peer review and any attached files.

Reviewer #1: No

Reviewer #2: No

---

## [Decision Letter · Decision Letter 1]

25 Jul 2022

Dear Drs Mary-Huard and Balding,

Thank you very much for submitting your Research Article entitled 'Fast and accurate joint inference of coancestry parameters for populations and/or individuals' to PLOS Genetics.

The manuscript was fully evaluated at the editorial level and by independent peer reviewers. The reviewers appreciated the attention to an important topic but identified some concerns that we ask you address in a revised manuscript. I have also made some minor comments in the appended file.

We therefore ask you to modify the manuscript according to the review recommendations. Your revisions should address the specific points made by each reviewer and myself..

[LINK]

Yours sincerely,

B.S Weir

Guest Editor

PLOS Genetics

Scott Williams

Section Editor

PLOS Genetics

Reviewer's Responses to Questions

**Comments to the Authors:**

Reviewer #1: Thanks for the careful revisions.

I think the paper can/should be accepted. My only suggestion would be that the authors think carefully about much of the work including admixture to include here. I think that work seems a bit premature, and it sounds like the authors are continuing to work on that topic. I think the current state of the admixture work is a bit raw, and I think having some of the work on admixture here may steal some thunder from the future paper.

In any case, I think that the abstract ought to make clear that this work is suited for populations whose history can be described accurately by a tree structure. It should be clear to the reader that the work has not yet encompassed situations where there is persistent and regularly occurring gene flow among populations, which is the case in many, many applications of Fst.

When I say above that the work is raw with regard to admixture, I mean that with regard to two points: 1. The paper is not clear about the process for inferring tree history with admixture. 2. The paper does not even consider the case common with many species, of recurrent ongoing gene flow among large proportions of populations.

Reviewer #2: This is an extensively revised version of a manuscript I reviewed some time ago. My comments were fairly positive and I only highlighted some relatively minor issues, which have been addressed in this revision.

The new version is very clearly written and I could only identify few additional and relatively minor issues:

The manuscript presents two new estimators of F_ST between pairs of populations corresponding to well known formulations, one based on correlations, F^W, and another based on mismatch, F^H.

While the focus of the Material and Methods sections provides equal attention to both estimators, the simulation results are completely focused on F^H; although a table with results is presented for F^W (Table 5), there is no reference in the text to F^W results. I realise that the authors have reasons for this (absence of a ‘baseline’ estimator for F^W), but presenting a table without referring to it seems odd. At the very least they need to mention that the precision of the F^W is lower than that of F^H. Also, at the beginning of this section (Simulation Study) they have to mention that they will focus on F^H and only present limited results for F^W and they should also explain why.

Another minor comment is that the new approach is based on several assumptions that are mentioned under Material and Methods but I think they should also be briefly mentioned in the discussion as many interested users may not read the methodological details but still need to be aware of the assumptions being made when they use the new estimators.

Finally, a minor comment regarding Figure 2: Unless I’m missing something, the two populations referred to at the end of the caption should be CD instead of BC.

**Have all data underlying the figures and results presented in the manuscript been provided?**

Reviewer #1: Yes

Reviewer #2: Yes

PLOS authors have the option to publish the peer review history of their article (what does this mean?). If published, this will include your full peer review and any attached files.

Reviewer #1: No

Reviewer #2: No

---

## [Editor Report · Decision Letter 2]

11 Oct 2022

Dear Dr Mary-Huard,

Thank you very much for submitting your Research Article entitled 'Fast and accurate joint inference of coancestry parameters for populations and/or individuals' to PLOS Genetics.

The manuscript was fully evaluated at the editorial level and by independent peer reviewers. The reviewers appreciated the attention to an important topic but identified some concerns that we ask you address in a revised manuscript.

We therefore ask you to modify the manuscript according to the review recommendations. Your revisions should address the specific points made by each reviewer.

Yours sincerely,

B.S Weir

Guest Editor

PLOS Genetics

Scott Williams

Section Editor

PLOS Genetics

PGENETICS-D-22-00076R2

Associate Editor Comments     The authors still include incorrect statements about cited publications [3] and [8]:

Line 8: [2] and [3] used the method of moments estimation, not sum-of-squares estimation.

*W*

Line 81: The statement “Following Weir-Hill [3] we define *F**ST *= Cor[*x_k_, y_k_*t |*p*) = 0 if *k */= *k**' *”

is not correct. The whole thrust of the Weir-Hill paper was to allow correlated allele frequencies

among populations.

Line 355-357: “[8] note that ‘it is not possible to find estimates for each [Fst]... when sampled populations have correlated sample allele frequencies.’ ” By changing *θ *to *F_ST_ *the present authors have given an inaccurate quote and a misrepresentation of the whole thrust of the Weir-Goudet paper. Weir and Goudet were clear to distinguish between probabilities of identity by descent (*θ*) and *F_ST_ *quantities (*β*). The rest of the quoted paragraph includes “it is possible to rank *β *values, and, we expect these to have the same ranking as their expected values *θ*” which changes the meaning of what the present authors imply. The actual expected values of the *β*’s are functions of different *θ*’s as shown in Table 3 of [8]. Weir and Goudet gave explicit closed-form estimators for *F_ST_ *and these estimators allow for the presence of inbreeding.

Some minor points include

Line 136: “For the proof, see” is missing a citation.Line 173: “*p *∈*/* (0*.*0*, *0*.*99)” in spite of authors’ claim in their response to reviews not to use (*...*) notation.

Line 284: “Greater divergence of non-AFR populations may reflect an out-of-Africa bottle- neck.” seems weak when compared to Line 304 “consistent with the origins of modern humans in Africa.”Line 333: “and hence FST estimates” should state which FST estimates are being used.

---

## [Editor Report · Decision Letter 3]

1 Dec 2022

Dear Dr Mary-Huard,

We are pleased to inform you that your manuscript entitled "Fast and accurate joint inference of coancestry parameters for populations and/or individuals" has been editorially accepted for publication in PLOS Genetics. Congratulations!

Yours sincerely,

B.S Weir

Guest Editor

PLOS Genetics

Scott Williams

Section Editor

PLOS Genetics

Comments from the reviewers (if applicable):

**Data Deposition**

http://datadryad.org/submit?journalID=pgenetics&manu=PGENETICS-D-22-00076R3

**Press Queries**

---

## [Editor Report · Acceptance letter]

16 Jan 2023

PGENETICS-D-22-00076R3 

Fast and accurate joint inference of coancestry parameters for populations and/or individuals 

Dear Dr Mary-Huard, 

We are pleased to inform you that your manuscript entitled "Fast and accurate joint inference of coancestry parameters for populations and/or individuals" has been formally accepted for publication in PLOS Genetics! Your manuscript is now with our production department and you will be notified of the publication date in due course.

With kind regards,

Bernadett Koltai

PLOS Genetics

On behalf of:
